# Vascular Remodeling Is a Crucial Event in the Early Phase of Hepatocarcinogenesis in Rodent Models for Liver Tumorigenesis

**DOI:** 10.3390/cells11142129

**Published:** 2022-07-06

**Authors:** Margaret Tulessin, Rim Sabrina Jahan Sarker, Joscha Griger, Thomas Leibing, Cyrill Geraud, Wilko Weichert, Katja Steiger, Carolin Mogler

**Affiliations:** 1Institute of Pathology, School of Medicine, Technical University of Munich (TUM), 81675 Munich, Germany; margaret.tulessin@tum.de (M.T.); sabrina.sarker@tum.de (R.S.J.S.); wilko.weichert@tum.de (W.W.); katja.steiger@tum.de (K.S.); 2Comparative Experimental Pathology, School of Medicine, Technical University of Munich (TUM), 81675 Munich, Germany; 3Institute of Molecular Oncology and Functional Genomics, School of Medicine, Technical University of Munich (TUM), 81675 Munich, Germany; joscha.griger@tum.de; 4Department of Dermatology, Venereology, and Allergology, University Medical Center and Medical Faculty Mannheim, Heidelberg University, 68167 Mannheim, Germany; thomas.leibing@medma.uni-heidelberg.de (T.L.); cyrill.geraud@umm.de (C.G.); 5Section of Clinical and Molecular Dermatology, University Medical Center and Medical Faculty Mannheim, Heidelberg University, 68167 Mannheim, Germany; 6European Center for Angioscience (ECAS), Medical Faculty Mannheim, Heidelberg University, 68167 Mannheim, Germany

**Keywords:** hepatocellular carcinoma, foci of cellular alteration, vessel analysis, vascular remodeling, image analysis, animal model

## Abstract

**Simple Summary:**

Hepatocellular carcinoma (HCC) is a highly vascularized tumor and remodeling of the tumor vasculature is one of the hallmarks of tumor progression. Mouse models are elegant tools to study the onset and progression of liver tumors. However, only few data exist on the vasculature and vascular remodeling processes especially in the early phase of hepatocarcinogenesis. The aim of this study was therefore to perform a comprehensive characterization and comparison of the vasculature in mouse models used for hepatocarcinogenesis studies. For this purpose, we characterized the preneoplastic foci of cellular alteration (FCA) and hepatocellular carcinoma (HCC) by using tissue-based techniques and computer-assisted analysis to better understand if and how vascular remodeling appears in rodent models for liver tumorigenesis. Our findings demonstrated crucial differences in the number and size of the vessels, degree of maturation and intratumoral localization of the vasculature in FCA and HCC, clearly indicating that vascular remodeling is an important step in the early phase of liver tumorigenesis of rodent models.

**Abstract:**

The investigation of hepatocarcinogenesis is a major field of interest in oncology research and rodent models are commonly used to unravel the pathophysiology of onset and progression of hepatocellular carcinoma. HCC is a highly vascularized tumor and vascular remodeling is one of the hallmarks of tumor progression. To date, only a few detailed data exist about the vasculature and vascular remodeling in rodent models used for hepatocarcinogenesis. In this study, the vasculature of HCC and the preneoplastic foci of alteration (FCA) of different mouse models with varying genetic backgrounds were comprehensively characterized by using immunohistochemistry (CD31, Collagen IV, αSMA, Desmin and LYVE1) and RNA in situ hybridization (VEGF-A). Computational image analysis was performed to evaluate selected parameters including microvessel density, pericyte coverage, vessel size, intratumoral vessel distribution and architecture using the Aperio ImageScope and Definiens software programs. HCC presented with a significantly lower number of vessels, but larger vessel size and increased coverage, leading to a higher degree of maturation, whereas FCA lesions presented with a higher microvessel density and a higher amount of smaller but more immature vessels. Our results clearly demonstrate that vascular remodeling is present and crucial in early stages of experimental hepatocarcinogenesis. In addition, our detailed characterization provides a strong basis for further angiogenesis studies in these experimental models.

## 1. Introduction

Primary liver tumors are the sixth-most commonly diagnosed cancer and third-leading cause of cancer death worldwide [1]. A total of 75–85% of those tumors are hepatocellular carcinoma (HCC) [2,3,4]. There are several predisposing factors for HCC, including chronic viral hepatitis, cirrhosis, alcohol abuse and non-alcoholic fatty liver disease (NAFLD) [5,6,7,8,9,10,11,12,13]. Independent of their etiology, HCC are highly vascularized neoplasms in which angiogenesis and vascular remodeling play an important role in tumor onset and progression [14,15,16]. This so called angiogenic switch is a hallmark in the development of liver cancer including the transformation into a fully arterialized vascular supply, which in turn further promotes tumor growth and disease progression [17]. This process is mainly driven by an imbalance of pro- and antiangiogenic factors caused by continuous tumor cell growth and subsequent development of hypoxia [18]. Regulated by prolyl hydroxylases (PHDs), oxygen sensors (including the family of hypoxia inducible factors such as HIF1alpha) enhance the production of the vascular endothelial growth factor (VEGF), which in turn leads to the formation of new blood vessels [19,20]. Not only VEGF but several other angiogenic molecules have been identified to promote this remodeling including insulin growth factor-2 (IGF-2), platelet-derived growth factor (PDGF) fibroblast growth factor (FGF), thrombospondin (TS) and the angiopoietin family (ANG) [14,17,21,22]. The development of tumor vessels is characterized by the formation of typically irregular-sized and -shaped vessels, with abnormal vascular branching pattern, tortuous properties, a high level of leakiness and partial coverage by pericytes with incomplete basal membrane [16,23]. Ultrastructural findings from electron microscopy studies further identified endothelial cell thickening, reduction or lack of fenestrations, formation of basement membranes, paucity of sinusoidal macrophages and a higher rate of small arterioles with smooth muscle in their walls [23]. In more recent comparative studies, both murine and human HCC presented with a robust loss of differentiation markers in liver sinusoidal endothelial markers (LSEC) [14] and the potency of endothelial cells to lose their polarity, resulting in stratification and protrusion into the vessel lumen [24]. Fully developed HCC tumor nodules in a Cre-inducible mouse model using the SV40 large T antigen were proven to establish a functional vasculature by cooption, remodeling, and angiogenic expansion of the preexisting sinusoidal liver vasculature with increasing signs of vascular immaturity during tumor progression [15]. The vasculature thus undergoes a subsequent transformation and remodeling with loss of the specifically differentiated morphology of healthy liver sinusoids and displaying characteristics of capillary and precapillary blood vessels [23]. Recent studies further discuss the involvement of (secreted) factors such as suppressors of cytokine signaling 2 (SOCS 2) and ATAD2 as a member of the ATP family in HCC progression, molecules well known from physiological liver regeneration [25,26].

Chemically induced or genetically engineered rodent models are widely used to investigate and modulate the process of hepatocarcinogenesis [27]. Both model types typically present with a wide range of histopathological diagnosis [28]. Among these, proliferative preneoplastic lesions such as foci of cellular alteration (FCA) and early (small) hepatocellular carcinoma (HCC) are common findings [29]. In particular, the former are rarely taken into account when performing studies on the development and progression of HCC [27] unless given evidence that FCA (especially clear cell, basophilic and eosinophilic FCA) are very likely to resemble dysplastic nodules (DN) in humans and progression to HCC has been demonstrated [30]. In particular in newly designed genetic mouse models, little is known on the vasculature in early stages of liver tumorigenesis; and to date, it is not yet fully clarified if the development and progression from FCA to HCC also include mechanisms of vascular remodeling, likewise observed in human hepatocarcinogenesis and some chemically induced models [31]. Our study tries to address these open questions performing a computer-assisted in-depth characterization of the vasculature with focus on proliferative lesions in early stages of tumorigenesis in GEMM.

## 2. Materials and Methods

Tissue collection: All tissue samples of mice were processed at the Comparative Experimental Pathology (CEP) at the Institute of Pathology, Technical University Munich (TUM). Animals were initially provided to our collaboration partners (J.G.) by the Welcome Trust Sanger Institute, Genome Campus, Hinxton, Cambridge, CB10 1SA, UK. Experiments were approved by the local ethical committees in both the UK and Germany (TV 55.2-2532.Vet_02-16-143, government of Oberbayern; year of approval included in number). Mice were all kept under standard laboratory conditions (12 h day/night cycle, water and standard diet ad libitum, no special diet). Only samples from animals originating from endpoint studies were included. Samples from animals with unclear/insufficient extent of genetic knockdown were excluded from this study. A total of 25 formalin-fixed paraffin-embedded (FFPE) blocks from KRAS [28], KRAS/adenosine kinase (Adk) [32] and KRAS/ nuclear factor IA (Nfia) [33] genetically engineered mice (GEMM) were included in this study. These mice have previously been extensively characterized and chosen for this study due to a high tumor burden including a high percentage of FCA and HCC tumor nodules [28]. No differences in terms of number or distribution of histological diagnosis were observed upon further genetic modification (ADk/Nfia) and all mice were used equally for this study.

The FFPE blocks were cut and tissue slides (2–3 µm) were stained with hematoxylin and eosin (H&E) according to standard protocols. Slides were then independently evaluated by two experienced liver and comparative pathologists (K.S. and C.M.) and diagnosed according to existing guidelines for diagnosis of proliferative liver lesions in rodents [29]. Lesions with definite morphological diagnosis of FCA (clear cell, basophilic and eosinophilic subtype) and HCC were then selected for this study.

*Immunohistochemistry:* The intralesional vasculature of FCA and HCC was characterized by immunohistochemistry including staining for vascular adhesion molecule CD31 [34] (1:100; DIA-310, Dianova, Hamburg, Germany), Collagen IV [35] (1:50; CL50451AP, Cedarlane, ON, Canada), smooth muscle actin (α-SMA) [24] (1:500; ab5694, Abcam, Cambridge, UK), LYVE1 [36] (1:7000; ab33682, Abcam, Cambridge, UK), and Desmin [15] (1:50; M0760, DAKO, Santa Clara, CA, USA) using standard protocols [37,38].

*RNAscope* (in situ hybridization (ISH) and immunohistochemistry): Levels of vascular endothelial growth factor A (VEGF-A) were assessed by RNAscope (RNAscope multiplex fluorescent reagent Kit v2 Assay, 323100-USM, ACD, Newark, NJ, USA) according to the manufacturer’s protocol.

Computer-assisted image analysis: Slides were scanned using the slide scanner Aperio AT2 (Leica Biosystems, Nussloch, Germany) at a magnification of 40×. Selected regions of interest (ROIs) were then manually annotated the on Aperio ImageScope software (Version 12.4.0.7018, Leica Biosystems). Analysis of intralesional vasculature was performed using computational approaches. Microvessel density (MVD) was assessed using Definiens Tissue Studio of CD31- and Collagen IV-stained vessels were analyzed by Definiens Architect (version XD 64 2.7, Definiens AG, Munich, Germany) using the algorithm ‘Marker Area Detection’. ROIs were transferred from ImageScope using a default feature in Definiens. Subgroups of blood vessel size were defined in accordance with published literature of vessel size definitions in rats [39] and adopted for computer-assisted evaluation referring to area (but not diameter). The vessel areas for subgroup analysis were measured by annotating the vessels and calculating the average of vessel area, in order to set the criteria. Size of the vessels was then used to automatically define 3 groups of vessels—small (<150 µm^2^), medium (150–500 µm^2^), large (>500 µm^2^) vessels. Computer-analyzed quantitative vessel data were normalized accordingly as a ratio, providing three values for each vessel group: stained vessel area per total lesion area, stained vessel number per total lesion area and average staining intensity of the lesion. Heatmaps were generated based on the density of the three vessel subgroups mentioned above, using Definiens Architect (version XD 64 2.7, Definiens AG, Munich, Germany). The heatmap evaluation was performed according to published literature [40], including colored-coded evaluation of density of marker expression (green color for the lowest density, yellow color for medium density, and red color for the highest density). The analysis was performed semi-quantitively, according to the highest hotspot locations (marked in red color) within the lesion (FCA or HCC) based on their computer-defined location (peripheral or central part of the lesion/intralesional). Each slide was evaluated by 2 independent investigators (R.S.J.S. and M.T.). The hotspot was referred to as the highest density of micro-vessel area (small/medium/large).

The staining for αSMA and Desmin were analyzed using the Aperio ImageScope software with the algorithm ‘Positive Pixel Count v9′ as previously described [12]. The default set of parameters of the algorithm was modified according to the stain contrast and intensity of the scanned images. The algorithm measured the intensity of the stain (brown signal) for the whole section. The total positive pixel was then normalized to the total area of the tissue section (pixel/mm^2^). The general expression of αSMA and Desmin was weak in both FCA and HCC lesions; therefore, pixels were counted automatically and intensity evaluated. LYVE1 expression was evaluated semi-quantitative and given as percentage (%) of vasculature/lesion stained for LYVE1.

For quantifying VEGF-A mRNA expression, an open-source image analysis software, ‘QuPath’ (version 0.2.3, Queen’s University Belfast, Ireland) was used [41]. The ROIs annotated on ImageScope were transferred as xml files onto QuPath using a software script developed by the QuPath developer. Firstly, cells were segmented using a modified ‘Cell detection’ algorithm. For probe (VEGF-A) detection, the ‘Subcellular detection’ algorithm was chosen and a detection threshold was adjusted interactively until all the probe dots are detected. The minimum and maximum spot size ranged from 0.5 to 3 µm^2^. Larger areas were considered as clusters of spots. The total number of subcellular spots of clusters for each ROI was counted. 

Statistical analysis: Statistical analyses were performed with GraphPad Prism (GraphPad Software 9.1.1. (223), Inc., San Diego, CA, United States). The cut off for statistical significance was *p* value ≤ 0.05. The selection of statistical test was performed according to the normal distribution tests (Shapiro–Wilk and Kolmogorov–Smirnov test). If groups were not normally distributed then a non-parametric test (Mann–Whitney U test) was performed, whereas an unpaired T-test was performed if groups were normally distributed. The heatmap analysis was statistically compared with Fisher’s eact test. Statistical supervision and guidance were performed by Dr. Katty Castillo and Birgit Waschulzik, Institute of Medical Informatics, Statistics and Epidemiology, Technical University of Munich (TUM), Munich, Germany.

## 3. Results

A total of 262 FCAs and 36 HCCs were identified by histological classification. Manual annotation for further computational analysis of H&E staining was performed (Figure 1A–D and Appendix A). Median FCA lesion in the cohort was 4.83 mm (ranging from 0.07–9.603 mm) and distributed multifocally up to 70 lesions per slide. The median HCC lesion size was 9.85 mm (range: 0.523–19.18 mm) and distributed mostly as one lesion per slide. Based on H&E morphology, FCA showed a homogenous vascular pattern with a predominant appearance of narrow vessels. The HCC sample, however, especially larger specimens, presented with a more inhomogeneous pattern including areas of narrow but also dilated and angled vessels surrounding tumor cell clusters. Necrosis was not observable within the smaller tumor nodules (FCA/HCC) but detectable in the larger HCC nodules. An overview of the various vascular patterns observed in H&E is shown in Appendix A. Immunohistochemical staining was performed against adhesion molecule CD31 to highlight endothelial cells (FCA in Figure 1E, HCC in Figure 1F, additional pictures of different patterns are in Appendix A) and Collagen IV to highlight vascular basal membrane components (FCA in Figure 1G, HCC in Figure 1H). In the subsequent computational analysis, the total number of vessels, assessed by CD31 and Collagen IV, was significantly higher in FCAs than in HCCs (Figure 2A,B). Analysis of staining intensities showed opposite results, where HCC presented with a stronger CD31 staining intensity but not Collagen IV (Figure 2C,D). No differences were observed in total area covered by intralesional vessels (Figure 2E,F).

For a detailed vessel analysis according to their size, a subgrouping of vessels was performed on CD31 (Figure 3A–D) and Collagen IV immunostaining (Figure 3E–H) dividing the vessels into small, medium and large (as described earlier in the Material and Methods section). Regarding the vessel area in CD31 staining, a significantly larger area in FCA was covered by small- and medium-sized vessels (Figure 4A,B), on the contrary HCC presented with a significant amount of large vessel areas (Figure 4C). In Collagen IV-stained vessel areas, and in CD31-stained vessels, FCA presented with a significantly larger area covered by small- and medium-sized vessels (Figure 4D,E), whereas no differences were observed in large vessel areas between FCA and HCC (Figure 4F). Consistent with the finding that a larger area was covered by small- and medium-sized vessels, the total number of small and medium vessels was similarly, significantly higher in FCA as compared to HCC, both in CD31- (Figure 4G–I) and Collagen IV- (Figure 4J,K) stained vessels. In addition, the number of large vessels was significantly higher in FCAs compared to that observed in HCCs, which was demonstrated by Collagen IV positive vessels (Figure 4L). On average staining intensity, HCC again showed a higher CD31 staining intensity in small, medium and large vessels (Figure 5A–C). In contrast, the intensity of Collagen IV staining (Figure 5D–F) did not show significant differences among vessel subgroups (a summary of the distribution of each vessel per subgroup per lesion is provided in Appendix A).

To obtain more detailed information about the intralesional distribution of vessels according to their size, a heatmap evaluation was additionally performed, based on CD31 and Collagen IV-stained and annotated slides (Figure 6A,B). The heatmap analysis of hotspots of CD31-stained vessels showed a strong correlation between vessel distribution and lesion in small- and medium-sized vessels (Figure 6), but not in large-sized vessels. Small- and medium-sized vessels were located more in the central part of the FCAs, compared to HCCs where these sizes of vessels located more in peripherally in the lesion. Regarding the heatmap analysis of Collagen IV-stained vessels, small vessels were predominantly located in the center (intralesional) of FCA, whereas Collagen IV-stained small vessels in the HCC lesions are mostly located peripherally in the lesion.

In the analysis of αSMA immunostaining, the number of pixels were significantly higher in HCC compared to FCA (Figure 7A,B), reflecting a higher coverage by pericytes. However, staining intensity did not differ between the two groups (Figure 7C). In LYVE1 staining, FCA and HCC vasculature expressed LYVE1 in a diffuse manner (Appendix A). In both types of lesions, only weak staining was expressed, predominantly less than in the sinusoids of surrounding healthy liver tissue was found (Appendix A). Interestingly, mostly larger FCAs and HCC, but not smaller FCAs, tended to have more LYVE1 expression, pronounced peripherally in the lesions; however, no significant differences were observed (Appendix A).

In Desmin staining analysis, neither pixel count nor intensity showed significant differences between FCA and HCC (Appendix A). To assess the amount of VEGF-A as a secreted protein, RNAscope was performed. Neither the number of positive spots and/or clusters (either counted separately or commonly) showed significant differences (Appendix A). A tabular summary of the main findings including differences in vessel number, vascular size, staining intensity as well as a simplified graphical presentation can be found in Appendix A.

## 4. Discussion

Hepatocellular carcinoma is among the leading cancer-related causes of death in the world [3]. To date, several mouse models for the study of hepatocarcinogenesis exist; however, available rodent models can be very diverse in terms of tumor development and histological subtypes heavily depending on their strain and biological background [28,42]. Often genetically engineered mice develop cancer without premalignant lesions or early tumors are not detected in the endpoint studies [28,43]. GEMM or chemically induced mouse models are widely used to study specific genes of interest or drug interactions but only few authors specifically address the role of the tumor vasculature [15].

Addressing one of the hallmarks of HCC development and progression, our study was designed to investigate the vasculature in genetic mouse models developing malignant and premalignant lesions to intensively study early vascular events. Our results comprehensively demonstrate very diverse vascular phenotypes in FCA and HCC. FCA presented with a higher number of small- and medium-sized vessels, higher levels of basal membrane components but a lower pericyte coverage. These findings reflect a vascular phenotype of immature small (capillary-like) vessels. Similar to the histological appearance, the vascular profile of FCA therefore closely resembles morphology and immunophenotype of human dysplastic nodules (DN) [34], further supporting the role of FCA as the direct murine counterpart to human DN [30]. In contrast, HCCs were characterized by a lower number of vessels, larger vessel size and a higher degree of pericyte coverage. These substantial changes in vascular architecture and composition define a complex maturation process towards a robust (arterial-like) vasculature capable of supporting proliferating tumor cells with oxygen and nutrients [18]. Similar observations of such an increased arterialized vasculature have been described in rat models of chemically induced hepatocarcinogenesis [31] and some extent also in mouse models for liver cancer [15]. The observed remodeling process from FCA to HCC did not include only vessel formation but also (spontaneous) vessel regression reflected by the decreasing numbers of endothelial cells and simultaneously stable or increasing levels of Collagen IV. The newly developed vessels are presumably functional in early HCC nodules as no tumor cell necrosis or hemorrhage could be detected in the smaller HCC nodules but only in larger [18]. The architectural changes in our mouse HCC mimic in part the “vessels encapsulating tumor clusters (VETC)” pattern commonly described in human HCC of predominantly macrotrabecular subtypes [44]. Furthermore, not only the size and degree of maturation but also the intralesional distribution of the vessels evolves in the development from FCA to HCC. FCA presented with small- and medium-sized vessels predominantly located in the central areas of the tumor nodules. However, in HCC, a clearly observable shift of small- and medium-sized vessels towards the outer region (periphery) could be observed supporting that vascular remodeling in HCC subsequently progresses to the periphery supporting infiltrative growth and progression of HCC. Similar vascular remodeling has been described in the progression of fully developed HCC to more advanced tumor stages, indicating a continuous vasculature adaption throughout the different stages of tumor development and differentiation [15]. As no significant differences were observed in the cellular levels of VEGF-A between our FCA and HCC, it remains to be speculated whether the involvement of VEGF-A might not yet play a leading role at this early time point or the observed remodeling might not be primarily driven by hypoxia, activation of oxygen sensors and subsequent enhancement of VEGF alone [19].

Our detailed analysis in summary provides clear evidence that the investigated mouse models reflect both morphologically and phenotypically the angiogenic switch in human hepatocarcinogenesis and can be therefore used as a suitable model to study vascular therapeutic approaches [45] or basic research questions in the early phase of tumor development to further unravel molecular pathways [27]. One limitation of this mouse model (precisely any mouse model) includes the general lack of availability of a more detailed classification of proliferative liver lesions compared to the actual WHO classification of human liver tumors [46]. The diagnosis of “HCC” in humans is subdivided in small, early and progressed HCC with precise diagnostic criteria for each category. In murine liver tumors, the INHAND criteria to date only provides the diagnosis of “HCC” without further subclassification [29]. This difference needs to be addressed when discussing different stages of HCC development in terms of sage of mouse models. The mouse models used in our study furthermore do not develop FCA or HCC on a cirrhotic background [28], a condition which should be carefully considered when choosing this model for specific questions on chronic liver diseases [47]. Vascular remodeling is broadly observed in physiological and pathological non-neoplastic liver conditions such as chronic inflammation, regeneration and liver fibrosis [21,23,34,36,48,49,50,51]. An emerging role of sinusoidal endothelial cells has recently also been described in the development and progression of non-alcoholic steatohepatitis by altered endocytosis of lipids by the endothelial cells and activation of Kupffer Cells [10,52]. Therefore, it should be taken into account that mouse models developing HCC on a cirrhotic background (such as the MDR2 mouse model [53,54]) should be investigated independently with regard to vascular remodeling and angiogenic switch.

## 5. Conclusions

Angiogenesis is a vital step in tumor onset and progression in HCC and its precursor lesions. In our research study, we contribute to closure in the gap of knowledge on tumor vasculature in the development of FCA to HCC in rodent hepatocarcinogenesis, by using an in-depth computational analysis of the tumor vasculature. Our results clearly demonstrate that vascular remodeling is present in early stages of liver tumorigenesis making these mouse models with a histological spectrum of FCA and HCC an attractive tool for angiogenesis research purposes.

## Figures and Tables

**Figure 1 cells-11-02129-f001:**
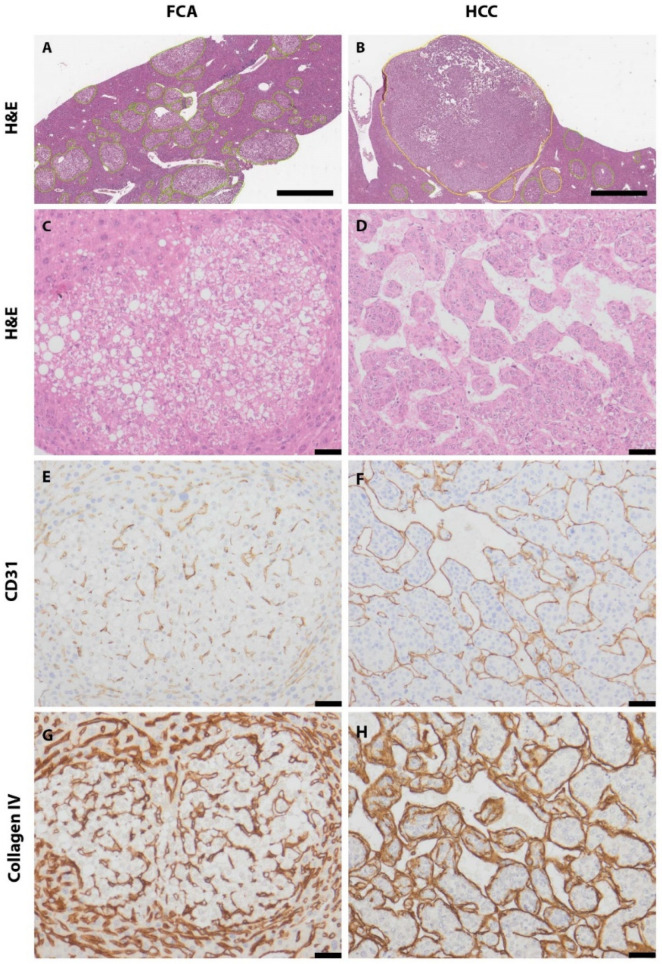
**Annotation and immunostaining of foci of cellular alteration (FCA) and hepatocellular carcinoma (HCC).** FCA annotated with green color in H&E (**A**); HCC annotated with yellow color in H&E (**B**); representative image of FCA (**C**) (H&E) and HCC (**D**) (H&E); CD31 immunostaining in FCA I and HCC (**E**,**F**). Collagen IV immunostaining in FCA (**G**) and HCC (**H**). Scale bar (**A**,**B**): 2 mm (magnification 1.5×). Scale bar (**C**–**F**): 50 µm (magnification 20×).

**Figure 2 cells-11-02129-f002:**
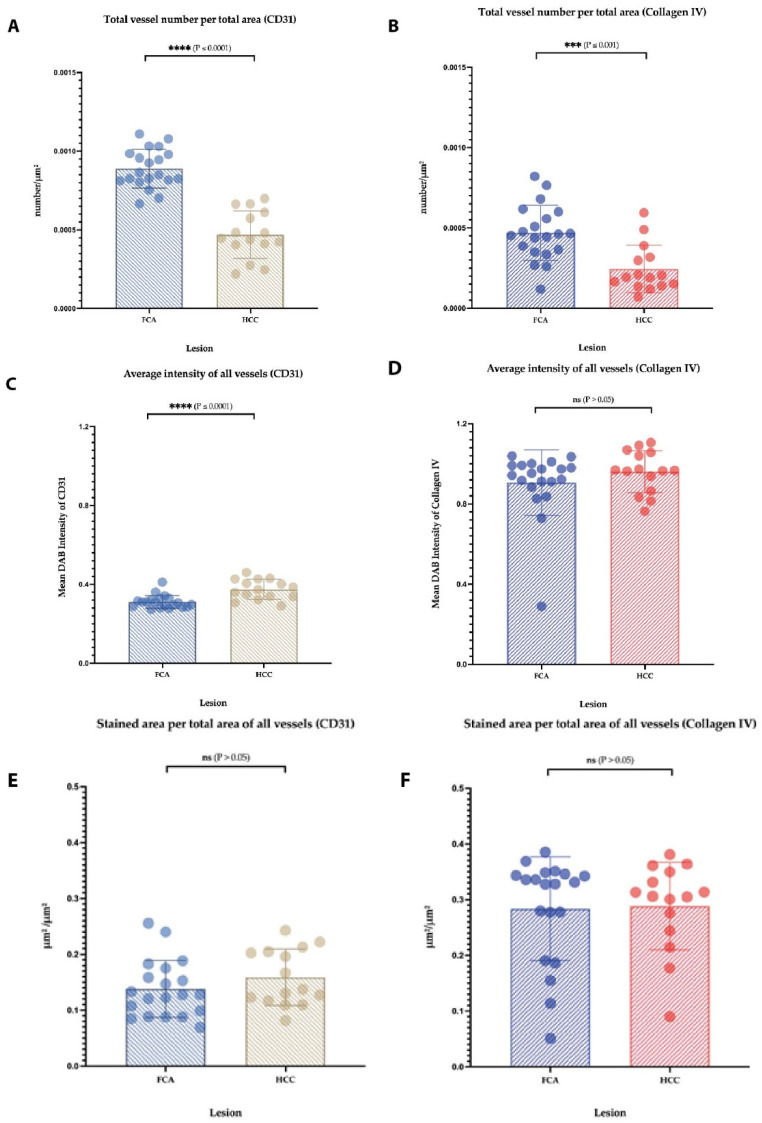
**Detailed vessel analysis by CD31 and Collagen IV in FCA and HCC.** Total number of vessels in FCA versus HCC by CD31 (**A**) and Collagen IV (**B**); average staining intensity of vessels by CD31 (**C**) and Collagen IV (**D**). No differences were observed in total vessel area: CD31 (**E**) and Collagen IV (**F**). Error bars show the mean and standard deviation for each lesion. *p*-values: Not statistically significant (ns) *p* value > 0.05; for statistical significance, accepted *** = *p* value ≤ 0.001, and **** = *p* value ≤ 0.0001.

**Figure 3 cells-11-02129-f003:**
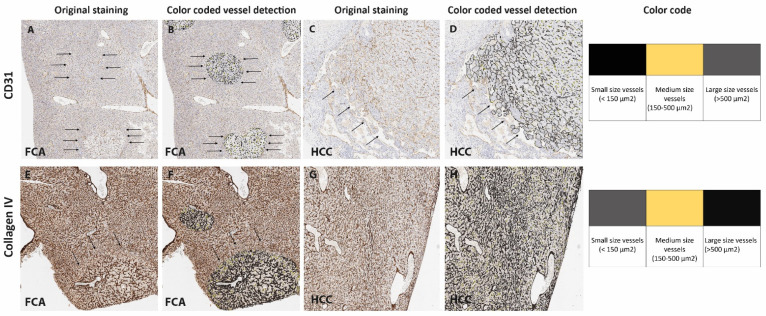
**Computer-assisted subgrouping of vessels.** CD31-based vessel subgrouping (**A**–**D**) for FCA (**A**) and CD31 (**B**) subgroups and HCC (**C**) and CD31 (**D**) subgroups. Collagen IV-based vessel subgrouping (**E**–**H**) for FI (**E**) and Collagen IV (**F**) subgroups and HCC (**G**) and Collagen IV (**H**) subgroups. Arrows mark the lesion in (**A**–**F**). Magnification 5×. Color coding of subgroups: (**A**,**D**) black-colored areas highlight small-sized vessels (<150 µm^2^), yellow-colored areas highlight medium-sized vessels (150–500 µm^2^), and grey-colored areas highlight large-sized vessels (>500 µm²). (**F**,**H**) Grey-colored areas highlight small-sized vessels (<150 µm^2^), yellow-colored areas highlight medium-sized vessels (150–500 µm^2^), and black-colored areas highlight large-sized vessels (>500 µm^2^).

**Figure 4 cells-11-02129-f004:**
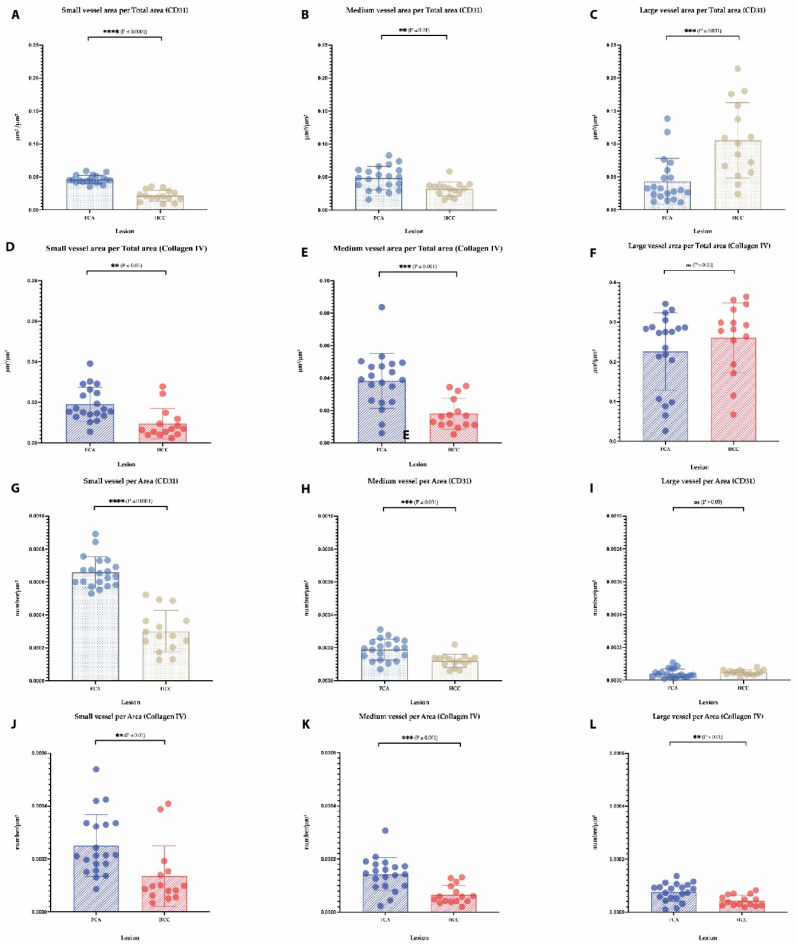
**Analysis of vessel area and vessel number per lesion.** Evaluation of subgrouped vessels per total area in CD31 (**A**–**C**) and Collagen IV staining (**D**,**E**,**F**). Subgrouped vessels per area in CD31 (**G**–**I**) and Collagen IV staining (**J**,**K**,**L**). Error bars show the mean and standard deviation for each lesion. *p*-values: not statistically significant (ns) *p* value > 0.05; for statistical significance, accepted *p* value ≤ 0.05., ** *p* value ≤ 0.01, *** *p* value ≤ 0.001, and **** *p* value ≤ 0.0001.

**Figure 5 cells-11-02129-f005:**
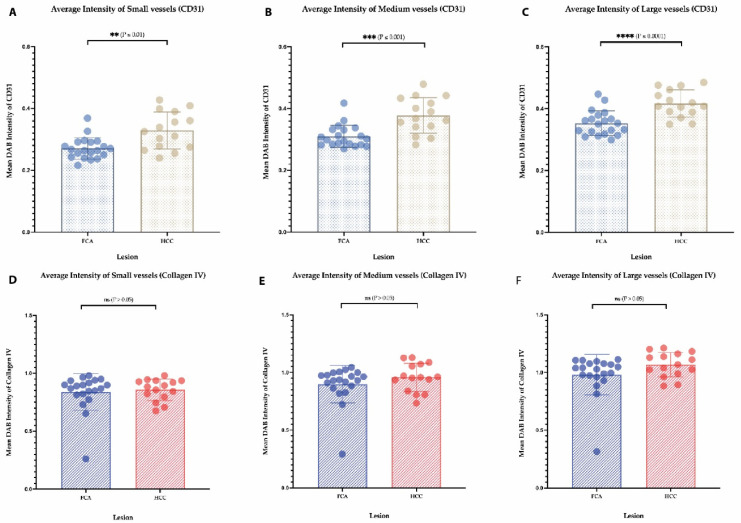
**Analysis of staining intensity per lesion of CD31- and Collagen IV-stained vessels (A–F).** Error bars indicate the mean and standard deviation for each lesion. *p*-values: Not statistically significant (ns) *p* value > 0.05; for statistical significance, accepted *p* value ≤ 0.05. ** *p* value ≤ 0.01, *** *p* value ≤ 0.001, and **** *p* value ≤ 0.0001.

**Figure 6 cells-11-02129-f006:**
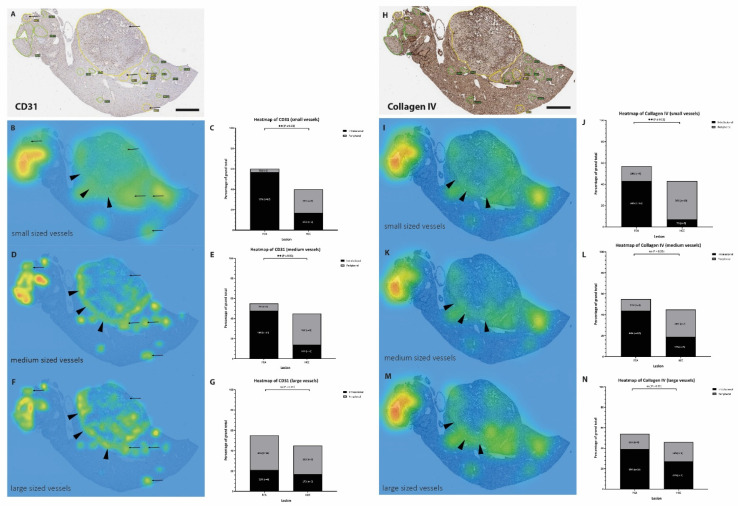
**Heatmap of vessel distribution according to size.** Annotated FCA (encircled green) and HCC (encircled yellow) in CD31 staining. (**A**) Distribution of the vessel according to their size in small-sized vessels (**B**,**C**), medium-sized vessels (**D**,**E**) and large-sized vessels (**F**,**G**) show a predominant location of small- and medium-sized vessels in the center (intralesional) of FCA, whereas the small- and medium-sized vessels in HCC mostly located at the periphery of HCC (arrowheads). In Collagen IV (**H**) small vessels located in the center (intralesional) of FCA but at the periphery of HCC, with shift towards the periphery in medium- and large-sized vessels (**I**–**N**). Color coding of heatmap: Green color indicates lowest density, yellow color indicates medium density and red color indicates highest density. Scale bar (**A**,**B**): 2 mm (magnification 1×). *p*-values: Not statistically significant (ns) *p* value > 0.05; for statistical significance, accepted *p* value ≤ 0.05. ** *p* value ≤ 0.01, N = number of hotspots identified in each slide and lesion for further in-depth analysis.

**Figure 7 cells-11-02129-f007:**
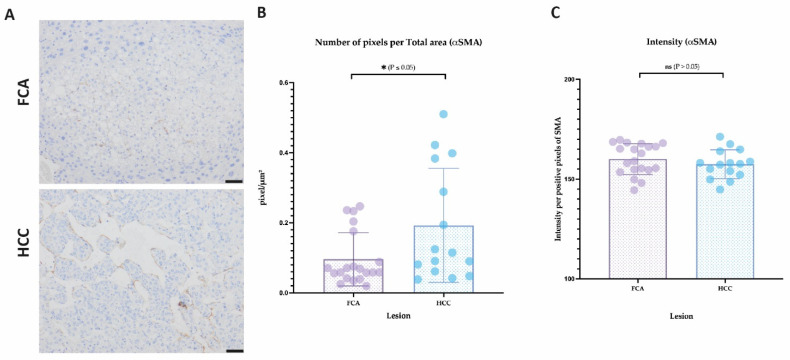
**Analysis of vessels per lesion for expression of smooth muscle actin (α-SMA).** (**A**) Immunostaining of α-SMA in FCA and HCC, (**B**) number of α-SMA positive pixel/total area and (**C**) intensity of α-SMA/positive pixel. for statistical significance, accepted *p* value: * *p* value ≤ 0.05. Scale bar: 50 µm (magnification 20×).

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
