# Peer review of "Vascular Remodeling Is a Crucial Event in the Early Phase of Hepatocarcinogenesis in Rodent Models for Liver Tumorigenesis"

_cells, 2022, doi:10.3390/cells11142129_

Round 1

Reviewer 1 Report

The authors present a primarily descriptive study regarding the transition of vascular phenotype from fca to hcc lesions. The data is interesting in that there is a general lack of information regarding the development of hcc from dysplastic lesions and this data certainly adds to it.  However, as it is primarily descriptive, and mechanistic data is not presented by the authors, the following recommendations are offered regarding suitability for publication.

1. Have the author's themselves or do they have access to any gene array/pcr data regarding vascular angiogenic transition in the lesions described? Can they provide any information on the genetic correlation to their phenotypes of smaller, immature vessels in FCA versus mature larger vessels in HCC. This would provide more supportive information regarding their IHC and heat map data

2. The authors should provide a bit more robust discussion in their introduction and conclusion regarding HCC and its development. It is acknowledged that overall data regarding HCC development is limited but there is relevant data regarding oxygen sensors, angiogenic molecules and hepatocyte regeneration that is likely relevant to this discussion that should be referenced and discussed.

Author Response

Point 1:

Have the author's themselves or do they have access to any gene array/pcr data regarding vascular angiogenic transition in the lesions described? Can they provide any information on the genetic correlation to their phenotypes of smaller, immature vessels in FCA versus mature larger vessels in HCC. This would provide more supportive information regarding their IHC and heat map data.

Response 1:

We thank the reviewer for his/her comment. We have already (before first submission of this manuscript) searched open data bases (such as TCGA) but could not find suitable genomic data on the vasculature in mouse FCA and HCC, especially none according to the size of the vessels. Like the reviewer suggested we fully agree that it would be ideal to have further genetic data (at least for discussion). Our collaborators indeed do have genomic data from tumor cell lines derived from either FCA and HCC tumor nodules from the mouse model used in our study. However, searching for 701 genes of vascular involvement in these cell-lined derived genomic data sets we did not found significant matches which could be used for our study results. We have therefore adressed this issue by expanding our discussion section accordingly to tyring to adress this lack of knowledge.

Point 2:

The authors should provide a bit more robust discussion in their introduction and conclusion regarding HCC and its development. It is acknowledged that overall data regarding HCC development is limited but there is relevant data regarding oxygen sensors, angiogenic molecules and hepatocyte regeneration that is likely relevant to this discussion that should be referenced and discussed.

Response 2:

According to the reviewer´s suggestion we have expanded our introduction & discussion and included more references concerning the above mentioned topics.

Reviewer 2 Report

The article entitled “Vascular remodeling is a crucial event in the early phase of 2 hepatocarcinogenesis in rodent models for liver tumorigenesis” is self-explanatory. This is a well-known condition and is accepted by most investigators in the allied fields. The authors are also aware of the fact and they have cited considerable numbers of references to validate that. The study presented here has been accomplished using the immunohistochemical technique. The study compared two clinical statuses of hepatocarcinogenesis; one is preneoplastic foci of alteration (FCA) and the other is hepatocellular carcinoma (HCC) lesions. FCA lesions presented with a higher microvessel density and a higher amount of smaller but more immature vessels, whereas, HCC presented with a significantly lower number of vessels, but larger vessel size. These findings induce the authors to state that vascular remodeling is present and crucial in the early stages of experimental hepatocarcinogenesis, the hallmark of the article.

Comments

1. The authors need to explain the characteristics of KRAS, KRAS/adenosine kinase (Adk) and KRAS/ nuclear factor 92 IA (Nfia) in genetically engineered mice (GEMM). Why these mice were used and what relation is prevalent with the use of these mice and the purpose of the study.

2. The relevance of CD31, Collagen IV, smooth muscle actin (-103 SMA), LYVE1, Desmin for staining FCA and HCC should be discussed. Only providing references is not enough.

3. What about a control group in this study?

4. The clinical relevance of the4 study should be properly cited. Does it prevail in early HCC as well? If this is similar in patients with liver cirrhosis progressing to HCC and patients directly progressing to HCC.

5. Remodeling of vasculature should be discussed in the context of other pathologies

Author Response

Please find the point by point response in the uploaded word document.

Reviewer 3 Report

As stated by the authors, the investigation of hepatocarcinogenesis is one major field of interest in the oncology research field, and rodent models are commonly used to understand the pathophysiology of onset and progression of liver cancer.

Among them, HCC is a highly vascularized tumor and vascular remodeling one of the hallmarks in tumor progression. In the presented study, the vasculature of HCC and preneoplastic foci of alteration (FCA) of different mouse models with varying genetic backgrounds was comprehensively characterized by using immunohistochemistry techniques (CD31, Collagen IV, aSMA, Desmin and LYVE1) and RNA in situ hybridization (VEGF-A). Thereafter, they performed computational image analysis to evaluate selected parameters including microvessel density, pericyte coverage, vessel size, intratumoral vessel distribution and architecture using software to obtain digital analytical data .

The findings in this study could be interesting but the following points should be improved for further consideration of this paper to be published in Cells.

1. The sample number of FCAs v HCC should be similar to perform fair comparison. At present, 262 vs 36 and recommended to have more Ns with HCC samples

2. Results from Figure 2, 4-5 should be summarized in a Table so that we can see the differences more clearly

3. Similar suggestion to the above statement, the author should not only describe the results but deeply find reasonable reason to explain why the differences are in FCAs and HCC. Neither in Discussion nor in result section or abstract we can find clear and detailed explanation (they explained some but not enough to cover all the experiments). In the abstract they mentioned that “HCC presented with a significant lower number of vessels, but larger vessel size and increased coverage leading to a higher degree of maturation, whereas FCA lesions presented with a higher microvessel density and a higher amount of smaller but more immature vessels”. It would make some sense but detailed comparison among all the results should be performed to better understand the phenomena they are presenting. At least Discussion should be more longer to explain what is new and what is confirmed regarding the previous studies. Additionally,  It will be better to link this analysis with human HCC how closely they are related in Discussion section to prove that their application can be useful for the human HCC development detection.

Author Response

Response to Reviewer 3 Comments

Point 1:

The sample number of FCAs v HCC should be similar to perform fair comparison. At present, 262 vs 36 and recommended to have more Ns with HCC samples.

Response 1:

Thank you for this suggestion. We explored this aspect. Previously in our 2020 Cancers paper (Steiger et al., Cancers 2020) we demonstrated that KRAS mice are predisposed to have more FCA and HCC lesions. In KRAS mice the fraction of FCA is more than at least 2 fold higher compared to HCC (67% vs 31%, respectively). This means is any number of mice, the prevalence of FCA will always be significantly higher than that of HCCs. That is the reason why the sample number of FCAs v HCC are not similar. In this cohort we had 25 KRAS mice, which in total had 262 FCA lesions and 36 HCC lesions. However, FCA in median size were much smaller than HCC (which reflects the nature of FCA as they further develop and progress into HCC), therefore all analysis were normalized to area and/or vessel number. Results were robust and statistically significant. Our investigations and analysis were all supervised by a professional statistician, we therefore think we can sufficiently use the data provided here with the given number of lesions.

Point 2:

Results from Figure 2, 4-5 should be summarized in a Table so that we can see the differences more clearly.

Response 2:

We thank the reviwer for this exzellent suggention – we have summarized our results in a (simplified) table and a cartoon. You can find both in our new supplementary figure 4.

Point 3:

Similar suggestion to the above statement, the author should not only describe the results but deeply find reasonable reason to explain why the differences are in FCAs and HCC. Neither in Discussion nor in result section or abstract we can find clear and detailed explanation (they explained some but not enough to cover all the experiments). In the abstract they mentioned that “HCC presented with a significant lower number of vessels, but larger vessel size and increased coverage leading to a higher degree of maturation, whereas FCA lesions presented with a higher microvessel density and a higher amount of smaller but more immature vessels”. It would make some sense but detailed comparison among all the results should be performed to better understand the phenomena they are presenting. At least Discussion should be more longer to explain what is new and what is confirmed regarding the previous studies. Additionally,  It will be better to link this analysis with human HCC how closely they are related in Discussion section to prove that their application can be useful for the human HCC development detection.

Response 3:

We thank the reviewer for her/his suggestions – we expanded the discussion section accordingly.

Round 2

Reviewer 1 Report

The authors have addressed the concerns raised in the first review and the manuscript is suitable for publication

Reviewer 2 Report

To the author

The queries provided by the Reviewers have been responded to by the authors.